# Aging Hallmarks and the Role of Oxidative Stress

**DOI:** 10.3390/antiox12030651

**Published:** 2023-03-06

**Authors:** Edio Maldonado, Sebastián Morales-Pison, Fabiola Urbina, Aldo Solari

**Affiliations:** 1Programa de Biología Celular y Molecular, Instituto de Ciencias Biomédicas, Facultad de Medicina, Universidad de Chile, Santiago 8380453, Chile; 2Centro de Oncología de Precisión (COP), Facultad de Medicina y Ciencias de la Salud, Universidad Mayor, Santiago 7560908, Chile

**Keywords:** oxidative stress, mitochondrial dysfunction, ROS, aging

## Abstract

Aging is a complex biological process accompanied by a progressive decline in the physical function of the organism and an increased risk of age-related chronic diseases such as cardiovascular diseases, cancer, and neurodegenerative diseases. Studies have established that there exist nine hallmarks of the aging process, including (i) telomere shortening, (ii) genomic instability, (iii) epigenetic modifications, (iv) mitochondrial dysfunction, (v) loss of proteostasis, (vi) dysregulated nutrient sensing, (vii) stem cell exhaustion, (viii) cellular senescence, and (ix) altered cellular communication. All these alterations have been linked to sustained systemic inflammation, and these mechanisms contribute to the aging process in timing not clearly determined yet. Nevertheless, mitochondrial dysfunction is one of the most important mechanisms contributing to the aging process. Mitochondria is the primary endogenous source of reactive oxygen species (ROS). During the aging process, there is a decline in ATP production and elevated ROS production together with a decline in the antioxidant defense. Elevated ROS levels can cause oxidative stress and severe damage to the cell, organelle membranes, DNA, lipids, and proteins. This damage contributes to the aging phenotype. In this review, we summarize recent advances in the mechanisms of aging with an emphasis on mitochondrial dysfunction and ROS production.

## 1. Introduction

Aging is a complex biological process influenced by genetic, epigenetic, environmental, and social factors. This process occurs along with a progressive decline in the physical function of the organism and an increased risk of age-related chronic diseases such as neurodegenerative disorders, cardiovascular disease, and cancer [1]. These diseases are the most prevalent worldwide and account for most morbidity and mortality [2]. Cardiovascular diseases are one of the most prevalent diseases of aging and account for 30% of all deaths worldwide [3]. Thus, the identification of biomarkers to diagnose and predict age-dependent risks has great value in preventing age-related diseases and improving the health status of the elderly. Whenever aging could be monitored, medical interventions could be performed before the early symptoms appear or before the appearance of the onset of chronic diseases.

Although at the population level, the overall risk of age-related diseases augments with chronological age, the age when the disease first emerges in a clinical form, the rates of disease development, and the age at death are highly variable between individuals. Therefore, two individuals with the same chronological age might have different health trajectories. The characterization of risk factors has proven useful for identifying individuals at high risk of developing specific age-related diseases. However, there needs to be a better method to distinguish between individuals at risk of developing age-related diseases that compromise their health status. Chronic inflammation is, at the moment, the only biomarker that predicts the development of multi-morbidity [4]. Therefore, identifying biomarkers that can discriminate, in early stages, individuals with different health outcomes with aging is a critical challenge in aging research.

Since aging is a complex biological process, many biomarkers would probably be needed to predict the physiological mechanisms of aging that leads to the multiplicity of adverse outcomes that typically occur in the elderly. Telomere shortening, occurring during cell replication, was initially considered predictive of age and age-related outcomes [5]. However, the most recent evidence indicates that an individual’s telomere shortening and age-associated effects are relatively low [5]. Telomere shortening inconsistently predicts the individuals’ clinical and functional outcomes, health, and lifespan. Therefore, other biomarkers are needed to predict age-related outcomes. 

The aging process results in several changes at the organism’s cellular and molecular levels. Studies have indicated that epigenetic changes are a significant component of the aging process [6]. Epigenetic changes comprise modulation of gene expression without any change in the genome sequence. Well-known epigenetic changes include histone modifications, DNA methylation, and non-coding RNA, and the changes in dynamic DNA methylation are the most found with the aging process. It has been well-established that there exist nine hallmarks of the aging process, which are: (i) telomere shortening, (ii) genomic instability, (iii) epigenetic modifications, (iv) mitochondrial dysfunction, (v) loss of proteostasis, (vi) deregulated nutrient sensing, (vii) stem cell exhaustion, (viii) cellular senescence, and (ix) altered cellular communication. All these alterations have been linked to sustained systemic inflammation [7]. The type of inflammation that occurs in the aging process is low-grade but persistent, damaging tissues and organs [7]. 

In this article, we will review different aspects of aging, focusing on the different biomarkers used to predict the physiological mechanics of the aging process and the cellular and molecular changes that occur. Important biomarkers include the systemic (plasmatic biomarkers) ones, which are easier to measure and monitor and reflect the deleterious consequences of the aging process, rather than its primary causes. We will also discuss the main aging hallmarks, especially the role of mitochondrial dysfunction and the production of ROS, of the aging process and the mechanisms underlying these alterations.

## 2. Epigenetic Alterations during the Aging Process

The concept of “epigenetics” was first introduced by C. Waddington in 1942 and refers to the modulation of gene expression with no changes in the genome sequence [8,9]. These changes occur at the histone modification and DNA methylation level, which were discovered at the end of the 20th century. Today, they have a well-established role in gene expression regulation [10,11]. 

### 2.1. Role of Histone Modification in Aging

Histone can bind DNA to compact it and accommodate its large size to the nucleus. The DNA–histone interaction is dynamic since the modifications at the lysine-rich tail domain of histones by small molecules alter the DNA–histone interactions. The histone modifications change the accessibility of that specific genomic area to modulate the activation, silencing, or rate of transcription of specific genes at that area [11,12]. Histone methylation usually occurs at the lysine (K) residues of histones H3 and H4 by adding methyl groups by the histone methyltransferase, using S-adenosylmethionine as the substrate to methylate lysine residues of histones [11,12]. This is one of the most important post-transcriptional modifications. In general, methylations H3K4me3 and H3K36me3 and acetylation H4K16ac are considered active marks occupying the actively transcribed gene regions in chromatin. While H3K9me3, H3K27me3, H3K56ac, and H4K20me2 are repressive marks associated with silenced gene expression and condensed chromatin [11,12]. On the other hand, histone acetylation occurs by the addition of acetyl groups to the K residues of histone tails and can be regulated by the balance between histone acetyltransferases (HATs) and histone deacetylases (HDACs) [11,12]. This modification can reduce the positive charge of the lysine residues, which will then inhibit the binding between histone tails and negatively charged DNA, leaving the DNA sites exposed to proteins of the transcription apparatus. Therefore, histone acetylation is considered an active mark. During the aging process, these modified sites can be subjected to changes, and a global reduction in heterochromatin can be observed [13,14,15]. Ranging from yeast to humans, aging is accompanied by a transition to more euchromatic states in specific regions that are normally heterochromatic, including telomeres and peri centromeres [15,16,17]. Distinct foci of heterochromatin might also be formed in the context of aging, thus mediating specific changes in gene expression [18]. 

### 2.2. DNA Methylation and the Aging Process 

A well-studied epigenetic change during the aging process is dynamic DNA methylation, which is found to be most tightly associated with the aging process [1,19,20,21]. Generally, dynamic changes in age-dependent DNA methylation include global hypomethylation and hypermethylation at specific genome regions (Figure 1) [21,22]. Many studies have demonstrated a very close correlation between DNA methylation, aging, and longevity [21,23,24]. Thus, DNA methylation status, evaluated by these predictors, can reflect the biological age of an individual that has a close association with the individual’s health status. These studies have allowed researchers to develop age predictors based on the correlation between DNA methylation and chronological age [25,26]. These predictors are named “epigenetic clocks”, and they consider the difference between DNA methylation age and chronological age (∆ age). These are promising tools to estimate disease risk and longevity potential in early life [27]. Several other clocks have been developed, most of them using different DNA methylation in cytosine guanine dinucleotides, named CpGs [28,29]. Comparing the different clocks suggests that increased epigenetic age relative to chronological age (known as age acceleration) can be associated with many adverse health outcomes [30]. The different epigenetic clocks exhibit different degrees of correlation with aging risk factors, indicating that they might be able to capture different aspects of the aging process.

DNA methylation refers to the transfer of a methyl group from S-adenosylmethionine to the fifth position of cytosine nucleotides (5mC), a process catalyzed by at least three DNA methyltransferases, including Dnmt1, Dnmt3a, and Dnmt3b [31,32]. Dnmt1 maintains the methylation patterns in the genome by replicating the hemimethylated CpG sites [32], while Dnmt3a and Dnmt3b are considered de novo DNA methylases [31]. On the other hand, DNA demethylation can be done by either passive or active mechanisms [33]. The active mechanism involves DNA demethylases, whereas the passive mechanism is achieved by Dnmt1 inhibition during cell replication [33,34]. 

In mammalian cells, 5mC mainly occurs at the genome sequence in the context of CpG dinucleotides, and nearly 70–80% of them are methylated in human somatic cells. Most unmethylated CpG sites are clustered in CpG islands, which are located at the promoter sequences of the genes [35,36]. DNA methylation near the transcription start site (TSS) is closely associated with the suppression of gene expression [37,38,39]. Evidence has shown that DNA methylation can inhibit the binding of transcription activation factor AP-2, preventing transcriptional activation and also can inhibit the recruitment of transcription-inhibiting factors (for example, AP-2 and McCP2) to the promoter regions [40,41,42]. On the contrary, there exist reports that DNA methylation of the gene body probably increases transcriptional activity [43,44]. Moreover, a recent report indicates that DNA methylation on the gene body can protect it from spurious transcripts guaranteeing the fidelity of the transcription process [45]. 

It is noteworthy to indicate that Dnmt1 has been found inside the mitochondria and interacts with the mitochondrial DNA in the matrix of some tissues, such as mouse embryonic fibroblasts and human colon carcinomas [46]. Further, Dnmt1 has been found inside the organelle and interacts with the mitochondrial DNA of human brain cells [47]. However, Dnmt1 is not the only Dnmt found in the mitochondria since studies have also detected the enzymes Dnmt3a and Dnmt3b [48,49]. These studies have found isoforms of these enzymes in the mitochondria, and it has been suggested that the presence of Dnmts and the differing levels of mitochondrial DNA methylation are tissue-specific [48]. Lopes, FC, postulates that the presence of these enzymes required for the epigenetic change in specific tissues might be functionally linked to the expression of specific mitochondrial genes [48]. Moreover, methylation of the mitochondrial master regulator PGC1α can be modulated by nuclear Dnmt1 and Dnmt3, and Dnmt1 activity can be inhibited by phosphorylation by AMP-activated protein kinase (AMPK) [50]. These results could indicate that differential methylation of nuclear genes and activity of Dnmt1 might also lead to changes in mitochondrial DNA methylation. 

Another enzyme type that could play a role in aging and age-related diseases are the ten eleven translocases (TET1-3). Methylation of cytosine in CpG dinucleotide forms 5mC, which is a stable and heritable epigenetic mark on the DNA. TET enzymes are responsible for the conversion of 5mC to 5-hydroxymethylcytosine (5hmC) during the demethylation process. Therefore, TET enzymes might be able to play a key role in maintaining the methylation balance at the CpG islands in the genome and potentially could influence the aging process. It has been found that the 5hmC content decreases with age in the peripheral blood T cells of humans [51]. TET2 expression rescues age-related decline in adult neurogenesis and enhances cognition in mice [52]. When mimicking an age-related condition in young adults by interfering with TET2 expression in the hippocampal neurogenic niche or adult neural stem cells, decreased neurogenesis and impaired learning and memory are seen [52]. Increasing the levels of TET2 in the hippocampal neurogenic niche of mature adults increases 5hmC, neutralizes the age-related decline in neurogenesis, and enhances learning and memory in these animals. Additionally, TET enzymes are able to regulate age-associated disorders such as cancer, cardiovascular disease, and stroke [53]. 

The association between DNA methylation and age has been studied for more than three decades. Wilson and Jones [54] observed a decrease in 5mC in normal-aged fibroblasts of hamsters, mice, and humans. From thereafter, hypomethylation has been observed in interspaced repetitive sequence (IRS) in several cell types from different tissues and organs [55,56]. Furthermore, over the human lifetime, the content of 5mC is higher in embryos and decreases in elderly individuals [57]. Further, the genome’s ribosomal DNA is hypermethylated in aged rats [58]. Several pieces of evidence have shown that the global reduced DNA hypomethylation can be caused by a down-regulated expression of Dnmts or an insufficient supply of folic acid in elderly individuals [59,60]. Several age-related diseases, such as neurodegenerative diseases, cardiovascular diseases, and cancer, show a very close association with global hypomethylation [35,59,61]. Additionally, abnormal methylation patterns in certain genes provide direct evidence of the close association between DNA methylation patterns and disease expression. As an example, the loss of DNA methylation in three CpG loci in the intron 1 of the triggering receptor expressed on myeloid cells 2 gene (TREM2, an Alzheimer’s disease susceptibility gene) results in a higher expression of TREM2 in leucocytes of Alzheimer’s patients [62]. All this evidence indicates the key role of DNA methylation in the process of aging.

Biological age is a concept to explain the variation in the biological status of individuals with the same chronological age [63]. There is increasing evidence that the difference between biological age (DNA methylation age) and chronological age (DNA methylation age-chronological age, ∆ age) is associated with age-related diseases. Several researchers have observed an increase in biological age in some age-related disorders such as cardiovascular disease, cancer, and Alzheimer’s disease [64,65]. Recent evidence has shown that ∆ age is able to predict mortality in later life [66,67]. For example, it has been shown that 5-year higher Hannum and Horvath ∆ ages are associated with 21 and 11% of greater mortality, respectively [66]. Another study by Zheng et al. [67] demonstrated that ∆ age can be used to predict cancer incidence and mortality. Many factors can accelerate the epigenetic clock, such as cumulative lifetime stress, drinking, and smoking, menopause in women, and obesity [21].

## 3. Role of Genomic Instability and Mutations in the Aging Process

Several studies have shown that the burden of DNA lesions is higher in older mammals than in younger ones. This seems to be a cause and not a consequence of aging. Currently, it is believed that genomic instability plays a key causal role in aging. The support for this idea comes from progeroid syndromes, in which inherited defects in DNA repair can increase the burden of DNA damage, which leads to accelerated aging of one or more organs in the individual. Additional evidence comes from the fact that accumulated DNA damage triggers cellular senescence and metabolic changes, promoting a decrease in tissue function and an increased susceptibility to age-related diseases. 

The defects in the DNA repair system are either inherited or emerge and accumulate with time. They can lead to genomic instability, which is characterized by a large number of mutations, chromosome aberration, and/or aneuploidy [68,69]. Several human diseases, defined as progeroid syndromes or diseases of accelerated aging, are characterized by hypersensitivity to genotoxins and defects in genome stability, and most of the genome instability disorders are characterized by accelerated aging of multiple organs [68]. 

Another line of evidence supporting the causal role of DNA damage in the aging process comes from oncology. Cancer patients treated with genotoxic agents age several decades faster than individuals not exposed to genotoxins [68,70]. It is well-known that radiation and chemotherapy produce a large amount of DNA damage, which is not specific only to tumor cells since the DNA of normal cells is also damaged and triggers apoptosis, senescence, or mutagenesis [68]. 

Further, there is evidence that DNA damage increases with age since the sources of damage also increase with age. One of the main sources of cumulative damage to biomolecules, including DNA, is oxidative stress (OS) caused by ROS, which modifies several biomolecules [68,71]. Further, it has been proposed that DNA repair capacity decreases with aging, leading to errors during the repair or replication of damaged DNA, producing mutations, including base substitutions, small insertions or deletions, and chromosome rearrangements [68,69]. Thus, somatic mutations are a stable molecular endpoint, indicating DNA damage and genomic instability [68]. 

## 4. The Role of Increased Cellular Senescence in Aging

Cellular senescence has been implicated as a key driver of aging and aging-related diseases. Senescent cells are characterized by a stable exit from the cell cycle and the loss of proliferative capacity even in the presence of nutrients and mitogenic stimuli [72,73]. Replicative senescence is caused by telomere shortening and induction of a DNA damage response, termed DDR, which facilitates DNA repair and arrests cell cycle progression until the DNA repair is completed [74]. Further, cellular senescence can be induced by other stressors, including epigenetic changes, genomic instability, reactive metabolites, mitochondrial dysfunction, OS, inactivation of certain tumor suppressor genes, oncogenic expression, and viral infections [72,73,74]. Intrinsic and extrinsic factors leading normal cells to senescent cells and their changes are shown in Figure 2. The cellular senescence program is initiated by p16/Rb and/or p21/p53 tumor suppressor pathways [73]. Of these, p16 and p21 are cyclin-dependent kinase inhibitors and tumor suppressors that act in a coordinated fashion and/or independently to arrest the cell cycle in G1 [75,76,77]. Expression of p16, which is known to be augmented in mammalian tissue with age, is a prominent marker of cellular senescence [78,79,80]. Moreover, senescent cells present DNA damage-associated features involving DNA segments with chromatin alterations reinforcing senescence (DNA-SCARS) and senescence-associated heterochromatin foci [81,82]. These chromatin alterations are potentially involved in determining cell fate decisions for cellular senescence [83,84]. It is thought that cellular senescence has evolved as an anti-tumor mechanism where the senescence-associated secretory phenotype (SASP) induced by oncogene-induced senescence can recruit immune cells to facilitate the removal of senescent cells [85,86,87]. Senescent cells are heterogeneous in nature and also pleiotropic in their function. These cells play an essential role in several normal physiological processes, including embryogenesis, tissue regeneration, cellular reprogramming, wound healing, immunosurveillance, and tumor suppression [72,73,88,89,90,91]. 

Senescent cells contribute to the pathology of many chronic diseases including cancer, osteoarthritis, diabetes, and neurodegenerative diseases for example Alzheimer’s disease [92,93]. Senescent cells accumulate with age in most tissues. The SASP factors, secreted by senescent cells, can act in proximal and distal fashions to induce secondary senescence, therefore propagating and augmenting the senescent cell burden [72,94,95]. Cellular senescence contributes to aging and also plays a causal role in many age-related diseases [72,73]. Accumulation of senescent cells frequently occurs at pathogenic sites of the main age-related chronic diseases such as neurodegenerative diseases, cardiovascular diseases, osteoporosis, renal disease, and liver cirrhosis [73]. Remarkably, the transplantation of a small number of senescent cells into young healthy animals recapitulates age-related impaired physical function [96,97]. The deleterious effects of senescent cells in aging and age-related diseases are most likely mediated by an increased SASP expression, which contains factors such as chemokines, TGF-β family members, and VEGF, which are inflammatory cytokines known to accelerate senescence accumulation by spreading senescence to neighbor cells [98,99]. Taken altogether, these pieces of evidence indicate that the accumulation of senescent cells with age and the deleterious effects of SASP are potent drivers of aging and age-related diseases. Both aging and age-related diseases shorten individuals’ lifespan and health span. Most importantly, the removal of senescent cells or the inhibition of the expression of SASP can delay or alleviate multiple age-related conditions underscoring the potential of targeting senescent cells [72,73]. 

## 5. Age-Related Sarcopenia

The term sarcopenia was introduced in 1988 by Irwin Rosenberg in reference to muscle wasting in older people [100]. The fundamental mechanisms contributing to age-related sarcopenia are not yet completely known, although advances have been made during the last two decades. Specifically, efforts have been focused on the identification of effectors contributing to sarcopenia, including impaired mitochondrial function, reducing the number of motor units, decline in number, and regenerative capacity of muscle stem cells, which correlates with cellular senescence in skeletal muscle. Further, another important factor affecting the aging process and contributing to age-related sarcopenia is low-grade systemic inflammation. In this section, we will briefly review the contribution of cellular senescence and the role of inflammation in age-related sarcopenia.

Sarcopenia is a syndrome of older people in the musculoskeletal system. It is currently recognized that there are multifactorial causes of age-related sarcopenia, such as impairments in neuromuscular function (loss of motor units innervating muscle), OS, mitochondrial dysfunction, the decline in anabolic hormones, systemic inflammation, and cellular senescence. This process detrimentally contributes to age-related sarcopenia via muscle stem cell dysfunction and the SASP factors [101]. Studies in mice and humans have indicated that targeting cellular senescence is a powerful tool to alleviate the process of age-related sarcopenia [101]. However, the underlying mechanisms by which cellular senescence contributes to age-related sarcopenia are not yet completely understood, and they need to be further investigated.

Recent evidence shows that chronic low-grade inflammation, an important factor affecting the aging process, can contribute to the loss of muscle mass, strength, and functionality, as it affects both muscle protein breakdown and synthesis [102]. Muscle protein metabolism is controlled by counterbalanced fluctuations in muscle protein breakdown and muscle protein synthesis. In older people, the balance between muscle protein breakdown and synthesis seems to be disturbed, leading to a progressive increase in the loss of skeletal muscle mass [102]. Many underlying factors can contribute to this condition; however, inflammation might play a key role in the regulation of muscle protein metabolism. Generally, the aging process is associated with a chronic state of slightly increased plasma levels of pro-inflammatory cytokines, such as tumor necrosis factor α (TNFα), interleukin 6 (IL-6), and C-reactive protein (CRP) among others [101]. This state is referred to as low-grade inflammation, and it is at least partially the result of a large number of cells leaving the cell cycle and entering the state of cellular senescence [101]. The senescent cells acquire the SASP, which can induce the production of pro-inflammatory cytokines (TNFα, IL-6, and overactivation of the NF-kB transcription factor) [103]. The data suggest that circulating concentrations of TNFα and IL-6 are significantly elevated in the sarcopenic elderly [104]. All these conditions suggest that inflammatory mediators affect muscle protein metabolism. Therefore, the challenge still remains to further uncover the specific molecular mechanisms by which inflammation affects muscle protein metabolism. Inflammation could induce muscle protein breakdown through several inflammatory signals. In general, proteolysis is regulated through four main pathways, including ubiquitin-proteasome, calpains, macro autophagy, and apoptosis [102]. 

## 6. Long Non-Coding RNAs in Aging

Long non-coding RNAs (lncRNAs) are a class of regulatory non-coding RNA with more than 200 nucleotides in length. They act as decoys, signals, guides, or scaffolds and impact gene expression at different modes of chromatin remodeling, transcriptional regulation, and post-transcriptional modifications [105,106]. LncRNAs show very little sequence or motif conservation among species or tissues, indicating high specificity and function diversity [105]. Based on their subcellular location, they can be divided into nuclear lncRNAs and cytoplasmic lncRNAs. The nuclear ones are implicated in transcription regulation, histone modification, chromatin remodeling, and early post-transcriptional regulation such as splicing [105]. On the other hand, cytoplasmic lncRNAs are associated with diverse post-transcriptional regulation, such as interactions with RNA binding proteins or RNAs to influence mRNA turnover, RNA location, stability, protein translation, and stability [107]. According to their genomic location, lncRNAs can be divided into sense, antisense, bidirectional, intron, intergenic, enhancer, and promoter [108,109]. 

Several lines of evidence demonstrate that lncRNAs can play a key role in cellular senescence and organism aging by regulating the cell cycle. It has been shown that lncRNA H19 is involved in cell proliferation, growth, and senescence, since imprinting deletion of insulin-like growth factor-2 (Igf-2)-H19 locus is involved in senescence [110]. Further, p21-associated non-coding RNA DNA damage activated (PANDA) has been reported that triggers DNA damage via p53, leading to G1 cell cycle arrest [111]. Additionally, several other lncRNAs are related to aging, such as Gadd7, 7SL, and FAL1, which affect the cell cycle [112,113,114]. LncRNAs participate in the progression of SASP and promote the secretion of inflammatory factors [115]. The lncRNA nuclear paraspeckle assembly transcript 1 (NEAT1) can be considered a novel inflammatory regulator since it can regulate paraspeckles and thus regulate senescence [116]. LET is another lncRNA that can stimulate the accumulation of nuclear factor 90 (NF90), which is able to inhibit the translation of several SASP factors [117,118]. The lncRNA IL7R can alleviate the inflammatory response induced by lipopolysaccharide, which suggests that this lncRNA might have a role in SASP production [119]. LncRNAs are also involved in telomere dynamics and telomere shortening. Telomer Repeat-containing RNA (TERRA) is a lncRNA transcribed at telomeric DNA sequences, and several studies have demonstrated that there is a connection between TERRA and aging [120,121]. It has been found that when TERRA was upregulated, telomere shortening was augmented, and replicative senescence enters prematurely in deficiency, centromeric instability and facial anomalies (ICF) syndrome type I appears. In addition, the telomeric RNA component (TERC), a 451-nucleotide lncRNA, is the core component of telomerase, providing a template for telomerase extension. Remarkably, the introduction of TERC rescued the premature aging phenotype of telomerase-deficient mice [122]. This suggests that TERC has a role in senescence and aging. On the other hand, lncRNAs can contribute to the recruitment of chromatin remodeling complexes during senescence and aging. The homeobox (HOX) antisense intergenic RNA (HOTAIR) plays a role in the recruitment and binding of chromatin remodeling complexes at the HOX sites, which leads to the retargeting of polycomb repressive factor 2 (PRC2) [123]. Recent studies have demonstrated that several lncRNAs, such as Air, H19, and TERRA, play a role in chromatin remodeling [108,124]. 

LncRNAs can also play roles in aging-related neurodegenerative diseases, such as Alzheimer’s disease, Parkinson’s disease, Huntington’s disease, and Amyotrophic Lateral Sclerosis. Several lncRNAs are involved in neural function, and their related RNA networks might be able to influence neurodegeneration [125,126]. The pathology of neurodegenerative diseases is mostly related to the accumulation of some proteins, and studies suggest that lncRNAs are associated with different aggregation events and thus with disease pathogenesis. Due to abnormal lncRNAs being associated with the aging process and with aging-related diseases, the aberrant expression of these molecules will have a profound impact on the pathogenesis of these diseases. We will not discuss further the relationship between lncRNAs and neurodegenerative diseases due to space constraints. Further details on the association between lncRNAs and neurodegenerative diseases can be found in a recent review by Ni et al. [106].

## 7. Stem Cell Exhaustion in Aging

The stem cell is an undifferentiated cell with the potential to divide to produce offspring cells that can continue as stem cells or become destined to differentiate. Stem cells are a continuous source of differentiated cells that make up the tissues and organs of plants and animals. There are two main types of stem cells, which are embryonic and adult. Adult stem cells are those that can differentiate in tissues.

Adult stem cells perform an essential function in maintaining tissue homeostasis and regeneration. Therefore, stem cell function’s quantitative and qualitative decline during life is known as stem cell exhaustion [126,127]. The process of stem cell exhaustion has been proposed as one of the main drivers of aging [126,127]. In support of this notion, age-associated phenotypes can be restored by inducting stem cell rejuvenation [128,129]. Adult stem cells play a key role in maintaining tissue homeostasis through repair and regeneration during the lifespan [130]. Stem cell exhaustion is observed in virtually all tissues and organs maintained by adult stem cells, such as bone, muscle, and forebrain [131,132]. Stem cell exhaustion is seen in human age-related diseases and in rare genetic disorders [127]. Further, stem cell exhaustion is one of the main hallmarks of aging, and many of the other cellular hallmarks of aging act as major drivers of the qualitative and quantitative changes which are observed in stem cell function during the aging process [127]. These hallmarks are often directly related to the function and characteristics of stem cells, including telomere shortening, cellular senescence, mitochondrial dysfunction, loss of proteostasis, genomic instability, epigenetic alterations, and altered cellular communication. 

The homeostasis of stem cells can be altered by several factors to decrease proliferation, which is one of the main features of stem cell aging [127]. Hematopoietic stem cells (HSCs) exhibit decreased rates of cell division in aged mice, indicating a general decline in cell-cycle activity [133]. The functional activity of HSCs can be diminished by replication stress caused by age-related cell-cycle defects such as DNA damage and chromosome aberrations, which leads to decreased blood production and impaired therapeutic potential in transplantation assays [134]. Increased levels of p16, a cell cycle regulator that inhibits cell-cycle progression, and the accumulation of DNA damage have been related to the decrease of stem cell populations during the aging process [127,133]. Conversely, in p16 deficient-aged HSCs, a rejuvenation and engraftment capacity can be observed [135]. However, a loss of quiescence and excessive proliferation cause premature exhaustion of cell progenitor stem cells due to the exhaustion of stem cell populations [127]. Further, genomic instability is a consequence of multiple types of stress or damage, and it is another important cause of the dysfunction and decline of adult stem cells during the aging process [127].

Several pieces of evidence in mammals suggest that epigenetic changes can drive adult stem cell aging, including HSCs, muscle stem cells (MuSCs), mesenchymal stem cells (MSCs), and satellite cells [127]. It has been shown that an increase in DNA damage and a decrease in the DNA repair capacity of HSCs during aging result in a progressive loss of HSCs [133,136]. In addition, there is a functional decline in HSCs in aged individuals due to the high levels of replication stress upon reentry into the cell cycle [134]. From these observations, the role of DNA damage and replication stress during HSC aging is clear; however, epigenetic dysregulation is also an important contributor to HSC exhaustion. Epigenetic profiling of young and old HSCs revealed changes in histone H3K4me3 levels across self-renewal genes and detected increased DNA methylation at genes involved in differentiation, leading to defects in differentiation during HSC aging [137]. Mouse HSCs can be promoted to proliferate by deletion of the deacetylase Sirtuin 6 (Sirt6), since Sirt6 represses the expression of Wnt target genes by interacting with the transcription factor LEF1 or by deacetylating H3K56Ac [138]. Epigenetic changes affecting MuSCs and MSCs leading to stem cell exhaustion have also been described, and these are reviewed in detail in reference [127].

## 8. Proteostasis Dysfunction in the Aging Process

Loss of protein homeostasis (proteostasis) is one of the hallmarks of age-related neurodegenerative diseases such as Alzheimer’s disease and Parkinson’s disease. A dysfunction in proteostasis causes the accumulation of unfolded or misfolded proteins, leading to the accumulation of protein aggregates in the cytoplasm and organelles. A complex proteostasis network, including chaperons, proteolytic machinery, and their regulators maintains proteostasis in healthy cells. That network is able to coordinate protein synthesis with protein folding. The proteostasis network operates to ensure that correctly folded polypeptides can be generated at the right time and in the right cellular location. Further, the proteostasis network allows the production of the right number of polypeptides to allow the stoichiometric assembly of oligomeric protein complexes. Further, it prevents protein misfolding and aggregation since misfolded proteins are removed by degradation mediated by the proteasome or by autophagy. The effectors of the proteostasis network are chaperons, which can ensure proper protein folding and cooperate with the degradation machinery. Proteostasis is altered in several pathological conditions, including mainly neurodegenerative diseases which are age-related. This indicates that the capacity of the proteostasis network can decline during the aging process. 

The inability of aged cells to maintain protein homeostasis is one of the main drivers of age-related cellular dysfunction and degenerative diseases [139,140,141]. In *C. elegans*, a model organism, studies have revealed that there is an extensive proteome remodeling with the development of a severe proteome imbalance during aging, which is featured by extensive changes in protein abundance [142,143,144,145]. These global changes are thought to be caused by the dysregulation of miRNA-mediated post-transcriptional gene regulation [143,145]. On the contrary, these changes were not observed in the tissues of aged mice since only small proteomics changes were observed [146]. Therefore, it seems to be that mammals devote more resources than *C. elegans* to maintain proteome balance and can limit age-dependent cell type and organelle-specific proteome changes [147]. 

One of the main features of the aging proteome is the loss of protein solubility with the concomitant accumulation of protein aggregates. In *C. elegans* a study of quantification of around 2100 aggregating proteins has revealed that proteins can maintain solubility at their physiological concentrations; however, they aggregate when exceeding that physiological concentration, which occurs with aging [143,148]. Low-abundance proteins tend to have higher aggregation propensities during aging compared with high-abundance proteins [146]. However, despite the greater solubility, high-abundance proteins predominantly contribute to a total load of protein aggregates [146]. The proteostasis network becomes increasingly burdened during aging due to the increasing load of misfolded and OS-damaged proteins [149]. This is particularly evident in non-dividing long-lived cells, such as neurons [150]. Additionally, in the human brain, the expression of ATP-dependent chaperones is repressed in aged individuals, which could further promote protein misfolding and aggregation [151]. Therefore, when the proteostasis network falls below a critical level, the aggregation-prone proteins can no longer be maintained in a soluble state producing protein aggregates [141]. Further, this threshold can fall in the presence of other stressors such as proteasome inhibition and mutations that can structurally destabilize proteins and promote misfolding of them [152,153]. Remarkably, stem cells are more resistant than differentiated cells to age-dependent proteostasis decline. For example, it has been demonstrated that embryonic stem cells have elevated levels of proteasome activity, which can degrade misfolded proteins [154]. Human pluripotent stem cells efficiently assemble the chaperonin complex named TRiC, which is made apparently by increasing the expression of the subunit CCT8, which can limit the complex assembly when it is present in sub stoichiometric amounts [155]. Asymmetrical stem cell division can also have a role in maintaining a balanced proteostasis since the differentiated cell can inherit the damaged protein [156,157,158]. 

## 9. Cell–Cell Communication Impairment during Aging

Optimal and coordinated intercellular communication between cells, tissues, and organs is fundamental for an organism’s homeostasis. Its dysregulation has been associated with aging and aging-related diseases. Many factors can contribute to altered cell–cell communication such as crosslinking of extracellular matrix molecules, such as collagen, creating a rigid cell environment, impeding cell–cell interactions, and cell migration inside the tissues. Further, regulatory and secreted effector molecules secreted into an immediate environment, which includes various SASP components released by senescent cells, hormones, systemic inflammatory molecules, and circulating regulatory RNAs, can also contribute to the altered intracellular communication and activity. 

During the aging process, intercellular communication alterations are mediated by gap junctions [159,160]. Gap junctions are intercellular channels that allow the traffic of second messenger molecules, ions, linear peptides, small metabolites, and small regulatory RNAs [161]. Gap junction channels are formed by two hexameric channels, named hemichannels or connexons, at the plasma membrane of each of the two adjacent cells. Each hemichannel is composed of six identical (homomeric channel) or different (heteromeric) members of the connexin (Cx) protein family [161]. There are twenty-one different connexin-encoding genes in the human genome, and the most broadly expressed and studied is connexin 43 (Cx43). Gap junction intercellular communication (GJIC) is essential in several tissues and organs to guarantee electrical and metabolic coupling. In the heart, the remodeling of Cx43 impairs GJIC and has been associated with defective electric conduction, producing cardiac disorders related to age-related cardiovascular diseases [161,162]. Despite the fact that Cx43 mRNA levels are similar between young and old animals, a decrease in Cx43 has been observed in aged hearts, together with the remodeling of the intercalated disc and slowed impulse propagation through the gap junction and thus the decline of the cardiac function [163,164,165,166]. Cx43 levels decrease due to the shortening of the UTRs of its mRNA, impairing translation [164]. In addition, a decrease of Cx43 expression is observed in the bones of aged animals, most likely due to the augmented OS in older animals [167,168]. Interestingly, the knockdown of Cx43 in osteocytes increases susceptibility to OS-induced cell death, suggesting that Cx43 has a protective role [169]. Depletion of Cx43 in osteocytes of young animals results in osteocyte apoptosis and decreases bone strength, resembling the observed phenotype at old age [170]. 

The importance of Cx43-mediated communication has also been shown in aging and aging-related processes in HSC. The Cx43 deficiency in HSC and progenitors (HSC/P) leads to a decrease in survival and cell cycle quiescence, which associates the lack of Cx43 with a propensity to senescence [171]. Indeed, the absence of Cx43 in HSC/P ends in the inability of the cells to drain ROS to the hematopoietic microenvironment, which leads to the accumulation of ROS within the cells, resulting in senescence [171]. The re-expression of Cx43 rescues the communication of HSC/P with bone marrow stromal cells, indicating that Cx43-mediated communication can exert a protective role against senescence and stem cell exhaustion, which indicates interconnectivity of the aging hallmarks [171]. 

Tunneling nanotubes (TNTs) are long and thin membrane structures that can mediate cell–cell communication. These structures are transient and dynamic in nature and are positive for F-actin, having a diameter of 50–700 nm and a length of up to 100 um [150]. TNTs allow the transfer of several molecules, such as proteins, signaling molecules, vesicles, small RNAs, prions, organelles, and viruses, between two connected cells [172]. Further, TNTs have been implicated in calcium flux and intercellular electrical coupling [173,174]. These structures have been reported in several cell types, including fibroblasts, immune, neural, epithelial, and cardiac cells [175,176,177,178,179]. TNTs have been implicated in several biological processes in these cells, including cancer progression, immune defense, pathogen spreading, development, transdifferentiation, stress resistance, and stress propagation [175,176,177,178,179]. Similar structures have been reported in vivo, associated with stem cell differentiation, cell migration, wound healing, embryogenesis, stress resistance, and neurovascular coupling [180,181]. A viral infection can trigger the formation of TNTs, OS, protein aggregates, prion-like proteins, and proinflammatory stimuli [178,182]. This suggests that TNT formation might be augmented during aging, in which many of these stimuli-phenomena are known to be increased. Additionally, senescent cells can communicate through TNTs or TNT-like structures, named cellular bridges, to directly transfer in vitro and in vivo proteins to neighboring cells [160]. This process might help immune surveillance of senescent cells by natural killer cells [183]. Furthermore, when the profile of proliferating and senescent human vascular smooth muscle cells are compared, cellular bridge formation and direct transfer occur preferentially in senescent cells [184]. Again, this indicates that there is an association between TNT formation and the process of aging. 

Another fundamental vector for cell–cell communication is extracellular vesicles (EVs), membrane-enclosed nanoparticles carrying information from the cell of origin to another cell at a distance and across the entire organism. EVs are loaded with several components, including nucleic acids, DNA, mRNA, miRNA, lncRNA, lipids, cytosolic and membrane proteins, and metabolites [182]. EVs are released into the extracellular space and can interact, depending on their cargo and composition, with recipient cells, triggering a response. After docking with the recipient cell, the EVs can directly activate receptors at the membrane triggering responses or releasing their cargo into the receptor cell. Several studies have shown that the amount and content of circulating EVs change with aging. It has been demonstrated that EVs secreted by senescent cells are functionally active in other cells [160]. They can be considered part of SASP, playing an essential role in alterations observed in aging [160]. It has been described that senescent cells secrete more EVs than young cells [185,186,187,188,189]. The increasing EV production in aged and senescent cells can be partly explained by the loss of proteostasis, another hallmark of aging. Furthermore, another data set suggests that EV secretion serves as a vehicle for removing unwanted molecules, such as misfolded proteins and damaged DNA, which could avoid an apoptotic-like death of senescent cells [190,191,192]. When EVs from healthy cells are compared with EVs from senescent cells, the EVs from senescent cells present changes in protein and miRNA contents to control the activity of other cells. As an example, there is an altered high level of IL-1β in circulating EVs in aged rats, which can be part of the inflammatory environment associated with aging [193]. 

Remarkably, EV senescent cells-derived can contribute to the induction of cellular senescence in normal cells through a mechanism mediated by IFN-induced transmembrane protein 3 (IFITM3), which is elevated in EVs from elderly donors compared with EVs isolated from young donors [194]. According to these observations, several studies have demonstrated that a small number of EVs derived from young animals can attenuate or even reverse aging-related effects [195,196,197,198,199,200,201]. For example, EVs isolated from the serum of young animals, but not the supernatant devoid of EVs, attenuated several inflammatory markers, such as IL-6 and IL-1β, in old recipient animals [195]. 

## 10. Mitochondrial Dysfunction Plays a Crucial Role in Aging

The mitochondria is the center of oxidative metabolism (ATP production) and the leading site of ROS production. ROS is considered to be mainly produced by electron leakage from complex I and complex III from electron transport chains, leading to a partial reduction of oxygen to form superoxide [202]. However, studies have demonstrated that, in muscle mitochondria collected from male Sprague Dawley rats, α-ketoglutarate dehydrogenase (KGDH), pyruvate dehydrogenase (PDH), and branched chain keto acid dehydrogenase (BCKDH) display higher natives for ROS production in comparison to complex I when Krebs cycle-linked metabolites are being oxidized [202]. Subsequently, superoxide is quickly dismutated to hydrogen peroxide by two dismutases, which are superoxide dismutase 2 (SOD2) in the mitochondrial matrix and superoxide dismutase 1 (SOD1) in the mitochondrial intermembrane space. Collectively, both superoxide and hydrogen peroxide generated in this process can be considered as mitochondrial ROS [203,204]. ROS can also be produced by exposure of cells to exogenous environmental agents, including light, ionizing and non-ionizing radiations, chemicals, drugs, and metals [204,205]. In addition, ROS can be produced by apoptosis, an inflammatory response involving the immune system. Elevated ROS levels can cause oxidative DNA damage and therefore mutagenesis [203,204]. Oxidative DNA damage can be induced in bystander cell populations. Bystander effects can be seen in cell populations neighboring or sharing media with damaged or stressed cells [206]. The in vitro signaling has been shown to be reminiscent of the inflammatory response mediated by COX-2-related pathways, which involves cytokines, growth factors, and membrane-permeable ROS [207]. ROS can be produced directly by damaged cells or indirectly via an inflammatory process and can pass to neighboring cells through different pathways such as passive diffusion, gap junctions, or active transport [208,209].

Both oxidative metabolism and ROS production are fundamental in health and the pathogenesis of many diseases. Redox signaling is important for regulating cell functions, but high concentrations of ROS are pathogenic and can cause severe damage to the mitochondria and other cellular components, which leads to aging (Figure 3). Mitochondrial dysfunction has been considered one of the primary mechanisms contributing to aging and functional decline [210,211]. 

The free radical theory of aging has been proposed in 1955 by Harman and links aging with OS [71]. In 1972, this theory was revised by the same author as the mitochondrial free radical theory of aging (MFRTA) [212]. The MFRTA states that mitochondrial dysfunction and consequent increased ROS production result in a cycle contributing to cellular damage and cell death. The ROS, which is produced during aerobic metabolism, interacts with cellular components, causing cumulative oxidative damage over time. The oxidative damage can cause damage to DNA, proteins, and lipids, which is associated with elevated ROS production, mitochondrial dysfunction, and ultimately cell senescence or death [213]. In this regard, the proximity between ROS production and mitochondrial DNA (mtDNA) favors the accumulation of OS-induced DNA damage in the organelle genome [214]. Elevated mitochondrial ROS production has been correlated with mitochondrial oxidative damage and a reduction of mitochondrial copy number [215]. The oxidative damage can cause an increased mutation rate of mtDNA in the brain, liver, and muscle fibers of elderly individuals [216]. Oxidated DNA products are a direct risk of genome stability, and the oxidative clustered DNA lesions, defined as two or more oxidative lesions, are present within 10 bp of each other [204]. DNA oxidative lessons occur mainly in guanines since they are the most easily oxidized. While DNA lesions are generally repaired, some of this oxidative damage can lead to DNA double-strand breaks (DSB), which are more difficult to resolve by the repair machinery before the damage interferes with DNA transcription and replication [204]. DSB is the most serious type of DNA damage and very few lesions of this type are sufficient to induce gene mutations, chromosomal aberrations, and cell transformation [216]. Telomeric DNA contains multiple repeats of TTAGGG, therefore is susceptible to ROS-induced damage [217,218]. Because telomeric DNA is shortened with each cell division up to a threshold length, the telomere-binding proteins can dissociate from it, resulting in uncapped telomeres and cellular growth arrest. On the other hand, considerable evidence suggests that OS plays a role in vitro in mammalian senescence [219]. In mice cells under culture, the onset of senescence was substantially delayed when the O2 level was reduced [220]. Similarly, when culturing bone marrow cells at low O2 levels, an increase was observed in plating efficiencies in this natural and rather hypoxic environment of stem cells [220,221]. 

Antioxidant defense efficiency also declines with age, which, together with increased ROS production, contributes to a manifestation of an OS state. This state can disturb enzyme activity by altering the structure and integrity. Consistently with this, the overexpression of antioxidant enzymes can decrease ROS production and protects DNA from its harmful effects, and this is associated with a prolonged lifespan in mice [222]. Furthermore, it has been found that long-lived mice strains contain higher levels of antioxidant enzymes, and they have reduced oxidative damage of proteins and lipids [223,224]. Further, the average lifespan of mice treated with antioxidant drugs increases up to 25% [225], and mice lacking the enzyme superoxide dismutase 1 (CuZnSOD1) exhibit a 30% decrease in life expectancy [226]. However, despite numerous studies supporting the ROS theory of aging, other recent studies have questioned a direct correlation between OS damage and lifespan. For example, using *C. elegans* as a model, Yang et al. found that mtDNA mutations do not affect overall ROS, despite an increase in mitochondrial superoxide level [227]. Several other similar studies can be found in the review by Giorgi et al. [214]. Nevertheless, there is still not enough evidence to undermine the MFRTA. These contradictory studies instead support a new theory named “mitohormesis”. According to this new theory, moderate levels of mitochondrial ROS might activate compensatory mechanisms, which can protect cellular organelles from the deleterious effects of ROS and consequently delay the appearance of an aging phenotype [228,229].

Mitochondria are not only the cell ATP-producing machinery but also constitute a latent source that triggers inflammation. When ROS or mtDNA leak out of the mitochondria, it causes activation of inflammasomes or cytosolic DNA sensors, respectively [230]. Mitochondrial dysfunction also can cause cell death when caspase activators, nucleases, or other harmful enzymes are released from the intermembrane space [230]. During aging, mitochondrial functions deteriorate due to multiple interlocked mechanisms, which include the accumulation of mtDNA mutations and altered (deficient) proteostasis, which leads to a destabilization of respiratory chain complexes, reduced turnover of the organelle, and changes in the mitochondrial dynamics. This compromises the activity of the mitochondria in ATP production, augments ROS production, and could trigger accidental permeabilization of the mitochondrial membrane due to lipid peroxidation, which can cause inflammation and cell death [231]. Thus, the mitochondrial function is primordial for health maintenance, and its progressive deterioration mainly contributes to the aging process.

Aging can also increase ER stress leading to mitochondrial dysfunction with decreased OXPHOS and ATP generation and increased ROS production in aged hearts [232]. Mitochondria are connected to the endoplasmic reticulum (ER) through mitochondria-associated membranes [233,234]. The dysfunctional mitochondria, in turn, can increase cardiac injury in aged hearts. The greater ER stress is indicated by increased expression of ER stress-related genes and proteins, including CHOP (C/EBP Homologous Protein) and cleaved ATF6 (Activating transcription factor 6) [232]. Moreover, aging can decrease the content of several mitochondrial proteins, especially subunits of electron transport complex I. However, the treatment of the aged heart with ER stress and mitochondrial dysfunction with 4-phenylbutyrate (4-PBA) can decrease ER stress and restore mitochondrial function [232]. Thus, ER stress is a key mechanism of age-induced mitochondrial dysfunction and can be improved by interventions.

### 10.1. ROS-Mediated Toxicity

The cells always try to keep the ROS levels required for their normal functioning; however, sometimes, there is an excess in ROS production, which is detrimental to cell functions. ROS excess can reduce the activity of enzymatic antioxidant defenses and also can decrease the activity of non-enzymatic proteins such as GSH. This affects the overall antioxidant defense system and makes it unable to eliminate the surplus free radicals. Excessive ROS are produced under hyperoxia and inflammatory conditions with a low or impaired antioxidant defense system, which finally alters the whole biological system [235]. For example, the oxidation of proteins and the production of nitrotyrosine (nitrosylation) can reduce the activity of enzymes and also can affect the production of growth factors, which leads to cellular dysfunction [236]. Lipid peroxidation is another damage produced by ROS excess, which damages membrane organelles through the activation of sphingomyelinase and ceramide release, which finally leads to cell death [237]. Moreover, the excess ROS causes oxidative DNA damage since ROS can react with the nucleic acids by attacking the nitrogenous bases and the sugar-phosphate backbones, inducing single and double-stranded DNA breaks, which are associated with premature aging when they are not completely repaired. 

An imbalance between free radicals and antioxidant defenses raises a tense-full situation in the living system known as OS. OS is a harmful process because it adversely affects the structure of cell membranes, lipids, proteins, lipoproteins, and DNA [235]. The excessive production of hydroxyl radicals and peroxynitrite leads to lipid peroxidation, therefore damaging cell membranes and lipoproteins. During the OS, the rate of lipid peroxidation increases very rapidly, affecting a large number of lipidic molecules [237]. Further, due to the peroxidative damage, proteins are affected by OS, undergoing structural modifications that produce a loss of or impairment of their enzymatic activities (Figure 3) [235]. Enzymes of the antioxidant defense system can be damaged, therefore producing a decrease in the cell’s antioxidant capacity. 

A prolonged OS causes DNA lesions, and the most common base affected is guanine in the DNA, and it produces 8-hydroxydeoxyguanosine (8-oxoG), which can bind thymidine rather than cytosine during the DNA replication process. The level of 8-oxoG is considered to be a pernicious DNA lesion, and it is responsible for the mutagenesis produced in the DNA [238,239]. Oxidative DNA damage can lead to a loss of both genetic and epigenetic information. The loss of epigenetic information is due to the damage in CpG island methylation asset in promoter regions of a gene [240], which is involved in promoter activity. Moreover, 8-oxoG acts as a tissue biomarker of OS. Thus, considering all these consequences, it can be concluded that OS can induce several diseases (acute and chronic), accelerating the cellular aging process and also causing pathologies, such as trauma and stroke [235]. Furthermore, OS can initiate several apoptotic signaling pathways due to increased ROS production or reduction in activities of the antioxidant enzymes, disruption of the intracellular redox homeostasis, and peroxidative damage in lipids, proteins, or DNA fragmentation [235]. The process of ROS-mediated toxicity leading to aging is displayed in Figure 3. 

### 10.2. Antioxidant Defenses

A decline in the antioxidant defenses, along with elevated ROS levels, is associated with the aging process. Normally, ROS balance is maintained by a cellular antioxidant defense system; however, in aging cells, there is a shift in ROS homeostasis toward a more OS status. The cells have several antioxidant systems to protect them from the deleterious effect of high ROS levels. These antioxidant systems are enzymatic and non-enzymatic, which allow the cells to deal with high ROS levels. Mainly, cells produce the enzymes glutathione peroxidase (GPX), GSH reductase, catalase, NADPH-quinone oxidoreductase-1 (NQO1), heme-oxygenase (HO-1), thioredoxin (Trx), and sulfiredoxin (Srx) [241,242]. Additionally, the transcription factor NRF2, which is a master regulator of cellular redox homeostasis, can upregulate the genes in charge of decreasing the OS. At basal levels, NRF2 is bound and repressed by KEAP1; however, in the presence of oxidative injury, KEAP1 can dissociate from the NRF2-KEAP1 complex and activates NRF2. Accumulated cytoplasmic NRF2 is then translocated to the nucleus and can bind to the antioxidant response element (ARE) in the promoter regions of genes that are involved in the antioxidant response [243]. In this regard, glutathione and thioredoxin are two key redox homeostatic antioxidant pathways controlled by NRF2. The cells also have non-enzymatic antioxidant mechanisms such as glutathione, uric acid, thioredoxin, pyridine nucleotides, vitamin C, vitamin E, retinoic acid, and metals such as zinc and selenium, which can provide efficient cellular protection against excessive OS [214]. Cells can use the antioxidants by either making use of cellular resources to produce endogenous antioxidants or by outsourcing the antioxidants from outside the cells from the diet. The main antioxidants from the diet include minerals, polyphenols, and carotenoids [244]. Taken altogether, these observations indicate that cells require a balanced ROS concentration for the intracellular signaling process; however, high ROS levels will produce OS in the cells.

Both aging and metabolic health degeneration have connections with increased OS [245]. ROS plays an important role in the aging process by triggering many age-related diseases through OS. Studies have found that cellular senescence in mesenchymal stem cells occurs by excessive ROS production due to a deficiency of complex I accompanied by depression of Ndufs6 [246]. Moreover, ROS also contributes to the aging of bone marrow-derived mesenchymal stem cells by aiding adipogenic differentiation and extinguishing osteogenic differentiation in mesenchymal stem cells. In this regard, it has been reported that the Indian Hedgehog (IHH), a ligand of the Hedgehog intracellular signaling pathway, can revive bone marrow-derived mesenchymal stem cells by routing intracellular ROS, which eventually could halt the STAT3, PI3K/AKT/NF-kB pathway, and down-regulate PI3K/AKT7mTOR pathways (involved in aging signaling) to terminate 4EBP1 and S6K phosphorylation [247]. 

## 11. Altered Nutrient Sensing in Aging

There are several types of nutrient-sensing molecules in mammals. The principal nutrient-sensing molecules in mammals include GCN2, mTOR, AMPK, and sirtuins [248]. Additionally, nutrient-sensing systems are regulated by several hormones, the most prominent being insulin and growth hormone and its secondary mediator, insulin-like growth factor 1 (IGF-1), which is produced in response to growth hormone by several cell types [248,249,250]. 

The insulin signaling pathway is the same as the one elicited by IGF-1, which informs cells of glucose availability. Therefore, insulin-IGF-1 signaling is known as the “insulin and IGF-1 signaling” (IIS) pathway. Importantly, the IIS signaling pathway is one of the most conserved aging-controlling pathways through evolution and has multiple downstream targets, including phosphoinositide 3-kinase (PI3K)/Akt, Ras/MAPK (mitogen-activated protein kinase), and mTOR (Figure 4) [249]. Therefore, aging is associated with altered nutrient-sensing mechanisms since, during the aging process, there are changes in the expression of many nutrient-sensing molecules, resulting in an altered metabolism. The same factors regulating growth also can regulate aging and manipulate the factors that decrease growth, resulting in decreased aging and prolonged lifespan [250]. Primarily, the aging process depends on genetic and epigenetic factors, but the environment to which the organism is exposed also affects the aging process and longevity. The IIS signaling pathway has been consistently demonstrated to influence aging in a broad spectrum of animals [248,251,252]. When this pathway is altered by dietary restriction (DR), which is to reduce food intake without causing malnutrition, there is a beneficial lifespan effect on organisms ranging from yeast to humans [250]. Furthermore, it has been demonstrated that there is a positive association between mutations of genes of the IIS signaling pathway and the extension of lifespan in humans [253,254]. The effect of DR on the IGF-1 signaling pathway is mediated through the class O of forkhead box (FoxO) transcription factors [255,256]. When IGF-1 binds to its receptor IGF1R and activates it, it in turn can activate the PI3K/Akt signaling cascade, the important main pathway for neuronal survival [251]. Once Akt is phosphorylated at Ser473, it becomes fully activated and can act on several downstream effector proteins, such as the FoxO transcription factors, which bind to the Forkhead response element and increase the expression of proapoptotic genes. Akt phosphorylates the Fox3a transcription factor, and it is retained in the cytoplasm, and it has been demonstrated that phospho-Fox3a is increased in old mice [251]. Further, phospho-Fox3a augments during OS, and this may be detrimental since this transcription factor is important for the expression of many antioxidant enzymes [257]. Additionally, one of the effectors of growth hormone is TOR, which is sensitive to many environmental cues, including nutrient availability and hormone actions [258], and integrates the signal from these factors to elicit several cellular processes. Specifically, mammalian TOR (mTOR) is considered to be one of the key regulators of aging and age-related processes [258,259]. The inhibition of this pathway confers protection against several age-associated diseases and extends the lifespan in several organisms [260]. Mammalian TOR promotes protein synthesis and negatively regulates autophagy [261,262], and it is a master regulator of growth and metabolism. Its downstream effectors are the ribosomal S6 kinase [S6K] pathway and the eukaryotic translation initiation factor 4E (eIF4E)-binding protein 1 (4E-BP1), a member of a family of translation repressor proteins [263]. TOR has a role in aging and longevity through protein synthesis regulation and autophagy [26,262]. Since protein synthesis is an expensive energy-demanding mechanism, it has been suggested that a reduction in protein synthesis might cause a decrease in energy consumption, therefore decreasing respiration and ROS production [264,265,266]. Indeed, pharmacological inhibition of mTOR by rapamycin extends the lifespan of mice (Figure 4) [267]. 

Lastly, emerging data highlights the link between mTOR and various processes relating to aging, such as mitochondrial dysfunction, maintenance of proteostasis, cellular senescence, and decline in stem cell function (Figure 4). Details of these links can be found in the review by Papadopoli et al. [267]. 

## 12. Growth Differentiation Factor 15 and Mitochondrial Dysfunction in the Aging Process 

GDF15 is a member of the transforming growth factor β-superfamily. This factor is an important stress-responsive secretory protein, and several studies indicate a possible link between mitochondrial dysfunction, GDF15, aging, and age-related diseases [268,269]. In different studies, a decreased mitochondrial function has been found in aged tissues and cells, and in studies of mitochondrial diseases caused by impaired respiratory chain function due to genetic mutations, the levels of GDF15 have been found to be elevated in animal models and patients [268,269]. 

Impaired mitochondrial respiration is characterized by decreased cellular ATP levels and increased NADH/NAD+ levels [269]. The decreased cellular ATP levels are deleterious to cell function. Although an increase in glycolysis can compensate for a decreased mitochondrial ATP generation, the high levels of NADH/NAD can suppress glycolysis. Therefore, decreasing the NADH/NAD+ levels promote cell function and survival despite impaired mitochondrial respiration. When mitochondrial respiration function is compromised, several different transcriptional programs can be triggered to maintain cellular homeostasis. A transcriptional study in cybrid cells (fusion of nucleated cells with cytoplasts) containing pathogenic mtDNA mutations revealed that activating transcription factor 4 (ATF4) and its target genes can be induced by mitochondrial dysfunction [270]. This plays a key role in mediating the integrated stress response (ISR) that can be induced by multiple stress stimuli, such as amino acid starvation, ER stress, viral infection, and heme deficiency [268,269]. Further, the transcriptional induction of downstream ISR genes has been demonstrated in cell cultures and patients with mitochondrial diseases [271,272]. A study using cybrid cells with mitochondrial myopathy, encephalopathy, lactic acidosis, and stroke-like episodes (MELAS) causing mtDNA mutations has shown that mitochondrial dysfunction induces GDF15 expression and its secretion, which can be further increased by severely impaired energy metabolism by treatment with exogenous lactate, which increases NADH/NAD+ levels [273]. Further, GDF15 expression is increased in cells exposed to respiratory chain inhibitors, both in animal models of mitochondrial diseases, and in the skeletal muscle of patients with mitochondrial diseases [271,272]. 

GDF15 gene expression is regulated by several transcription factors, which include the stress-responsive transcription factor p53, which mediates GDF15 increase in response to DNA damage [274]. However, it has not yet been demonstrated whether p53 directly regulates GDF15 expression in impaired mitochondrial respiration. The ATF4 knockdown suppresses GDF15 transcriptional activation induced by mitochondrial translation inhibition [271]. In addition, CHOP, a downstream transcription factor of ATF4 and p38 MAPK target, directly augments GDF15 expression when mitochondrial translation is impaired [275]. However, the effects of GDF15 induction on mitochondrial dysfunction and impaired energy still remain unknown. There is enough evidence to support the utility of this factor as a robust, helpful biomarker for impaired energy metabolism in cells with mitochondrial dysfunction. GDF15 circulates in fluids such as blood and cerebrospinal fluid under physiological conditions; however, serum GDF15 levels are significantly increased in patients with mitochondrial and aging-related diseases (Figure 5) [269,273]. The increased levels of circulating GDF15 are most likely the consequence of lesion tissues having a mitochondrial dysfunction.

## 13. Aging Biomarkers

It is helpful to develop ways to measure how fast an individual is aging. This information on biological age could be used to predict various health outcomes, such as cognitive and physical function, morbidity, and mortality. The characterization of risk factors has proven helpful for identifying individuals at risk of developing determined diseases. However, no adequate methods exist so far for identifying individuals at risk of developing specific diseases. Chronic inflammation is a biomarker that predicts the development of multimorbidity [4]. Therefore, developing biomarkers that can discriminate, at early stages, individuals with different health trajectories with aging is needed. One specific biomarker that does not meet the clinical precision requirement or lacks large-scale evidence is the case of interleukin 6 (IL-6), one of the main drivers of age-related increased susceptibility to chronic diseases, multimorbidity, and decline of physical functions [276]. 

### 13.1. OMICS in Aging

Several attempts have been made to unravel aging mechanisms via OMICS approaches. Modern technologies have opened an array of opportunities to quantify aging not only with a few numbers of well-studied and picked biomarkers but also with analyses of many thousands of genes, gene products, epigenetic modifications, and/or metabolites.

Genome-wide association studies (GWAS) are powerful approaches to examine the genetic architecture of polygenic traits in the genomes of individuals. Analyses of genomes in the elderly provide insights into the genetic predisposition of exceptional longevity [277,278,279], suggesting the existence of specific gene variants that centenarians should share. However, only the apolipoprotein E (APOE), the longevity locus on chromosome 5q33.3 (5q33.3 loci), and the stress-induced transcription factor forkhead box O3 (Fox3) are consistently associated with longer lifespan heritability across several studies that have been performed [279]. In a recent study of GWAS, 12 candidate loci associated with health span were found. Among them, three single nucleotide polymorphisms located at or near the cyclin-dependent kinase inhibitor (CDKN2B) gene, apolipoprotein A (LPA) gene, and the leukocyte antigen DQA1 gene (HLA-DQA1) were described [280].

Regarding the aging process, one of the best approaches to determine aging is the estimation of epigenetic clock signatures that were developed by examining DNA methylation across a large number of DNA sites to single out these that in a weighted average reproducibility predict chronological age [26]. Cytosine methylation at specific loci containing CpG dinucleotides changes differently with age, becoming hyper- or hypomethylated in different genomic locations. A significant breakthrough in biological measurements of age came with the development of a well-known epigenetic clock based on a correlation between the chronological age and methylation status of selected CpG loci [26]. Other epigenetic modifications include histone methylation and acetylation which are highly dynamic processes that occur over the human lifespan and are influenced by environmental and genetic factors [281].

Transcriptomic studies of human whole-blood gene expression revealed a group of about 1500 genes differentially expressed with chronological age [282]. The identified genes were consistent with well-known aging mechanisms including transcriptional and translational dysregulation, ribosome and mitochondria dysfunction, DNA damage accumulation, immune senescence, and altered metabolism [282]. Therefore, transcriptome significatively changes during the aging process, and the studies that have been carried out indicate that changes in the transcriptome are highly tissue-specific [283,284,285]. On the other hand, microRNAs (miRNAs) are small non-coding RNA molecules that participate in the regulation of gene expression by base pairing to complementary mRNA sequences, triggering mRNA cleavage and translational inhibition. These small RNAs can be found circulating in blood serum, within exosomes/microsomes, or bound to proteins or lipoprotein factors [286]. In humans, several mRNAs have been identified to be up- or down-regulated during aging in whole blood or serum, and the first microRNA age prediction model based on whole blood microRNA expression profiling has been established [287].

### 13.2. Proteomic Studies in Aging

The protein composition of cells, body fluids, and tissues changes with age. These changes can provide information about complex biological processes since proteins often directly regulate biological pathways. Particularly, blood contains proteins from nearly every cell and tissue, which can be analyzed to identify biomarkers and gain information about disease biology. Therefore, plasma protein changes reflect diverse aspects of the aging of different cells and tissues. The evidence that plasma proteins can be used to study aging comes from heterochronic parabiosis experiments [131,132]. This state connects by surgery the circulatory system of young and old mice. These studies have shown that multiple tissues including the heart, muscle, liver, pancreas, kidney, bone, and brain can be rejuvenated in old mice. Indeed, plasma from old mice is sufficient for brain aging in young mice, and young plasma can revert brain aging. These results indicate that plasma contains key regulators of age. Proteins are often direct regulators of cellular pathways, therefore should offer more informative links to age-related pathology and aging. Proteins are particularly attractive because they are direct biological effectors, and their dynamic changes with disease or aging are complex and diverse. Highly robust and clinically useful biomarkers will emerge after repeated studies in genetically and geographically diverse cohorts to predict biological age. Regarding proteins, many factors can influence the aging proteome such as post-translation modification, protein folding, protein aggregation, protein turnover, miRNA presence, and proteins shed into biofluids for easy diagnosis [126]. 

Interestingly, the association between chronological age and abundance of 1301 plasma proteins in a representative population sample of 997 participants (between 21 and 102 years old) of an Italian-based study (InCHIANT) was performed [288]. DNA methylation as an underlying epigenetic mechanism for age-related changes in protein abundance was included in this study. The study showed that for some proteins, DNA methylation could partly explain the observed age-associated changes. There was significant enrichment of age-associated methylation CpG sites within 10 kb of genes encoding for age-associated proteins when compared with genes coding for other unrelated proteins. The most significant association was observed for ectonucleotide pyrophosphatase/phosphodiesterase 7 (ENPP7), a member of the same family implicated in phospholipid and cholesterol metabolism with a CpG site cg 15739835 [288]. There is a negative association between cg15739835 methylation with age, reflecting the lower degree of methylation at an older age at this CpG site. Further, a positive association of ENPP7 abundance with age was found, and this association is attenuated when adjusted for cg15739835 methylation [288]. This indicates that the higher abundance of ENPPB7 in the elderly can be explained by the low methylation of its encoding gene. 

Of the age-associated proteins, 33.5% and 45.3% were associated with all-cause mortality and multimorbidity, respectively [288]. There was an enrichment of proteins associated with inflammation (TNF-activated receptor, chemokine receptor binding), regulation of gene expression (DNA methylation, meiosis, epigenetic regulation), and extracellular matrix (activation of matrix metalloproteinases, basement membrane, extracellular matrix organization). The most frequent associations among proteins were interleukin 10 (IL-10) signaling, ephrin receptor 9 (EPH9) signaling, and chemokines. Further, senescent cells and the SASP, which is characterized by enrichment in inflammatory cytokines [91], are potential sources of circulating pro-aging factors in the plasma. Indeed, it has been demonstrated that core SASP factors are enriched in the plasma of healthy aging individuals [289]. Among the 175 mortality-associated proteins that increased with age, 13 core SASP factors were identified, including three of the four top core SASP proteins, which are GDF15, matrix metallopeptidase 1 (MMT1), and stanniocalcin (STC1). Further, other classical SASP factors were found, such as insulin-like growth factor 1 binding protein 2–4 (IGFBP2), tissue inhibitor of metalloproteinases 1 and 2 (TIMP1 and 2), and IL-6. Plasma proteins most predictive of all-cause mortality were GDF15 and Beta-2-microglobulin (B2M), whereas a multiprotein model is composed of eight predictors, including age and proteins, namely IGFBP2, macrophage metalloelastase 12 (MMP12), epidermal growth factor receptor (EGFR), nuclear pore complex protein 8 NPP8, GDF15, peptidase inhibitor 3 (PI3), and growth hormone receptor (GHR).

A recent study of data mining of human plasma proteins has generated a highly predictive aging clock, which can reflect different aspects of aging [290]. The analysis identified 529 plasma proteins that have been reported to change their expression level with age by two or more studies [290,291]. When the q-value and age coefficient in a plasma proteomic dataset derived from 4263 healthy individuals (age range 18–95 years old) was analyzed, it was found that 476 proteins significatively changed their expression. Of these, 115 had a decreased expression level with age, while 361 had an increased expression level with age. The six proteins with the lowest q-value are chorionic gonadotrophin subunit alpha (CGA), follicle-stimulating hormone subunit beta (FSHB), sequestosome 1 (SQSTM1), GDF15, motilin (MLN), ret proto-oncogene (RET), and pleiotrophin (PTN). Among these, at least 64 proteins increase or decrease life span when altered in normal animal models, and 35 of these proteins affect lifespan in a vertebrate model. Furthermore, the number of lifespan regulatory proteins can be expanded to 108 when disease models, stress models, and models containing multiple genetic alterations are included. Most importantly, nine proteins were found that significantly extend lifespan when altered in normal, non-diseased mice or fish. These proteins include AKT2 kinase, growth differentiation factor 11 (GDF11), GDF15, nicotinamide phosphoribosyl transferase (NAMPT), pappalysin 1 (PAPPA), GHR, plasminogen activator urokinase (PLAU), phosphatase and tensin homolog (PTEN), and SHC-transforming protein 1 (SHC1). 

GDF15 is one of the best-associated proteins with aging. It is considered a stress-responsive cytokine induced by mitochondrial dysfunction, cellular stress, inflammation, or mitochondrial unfolded protein response (UPR), with positive effects on the health and life span of model organisms. GDF15 expression is low in healthy and young individuals; however, its levels dramatically augment in chronic or acute illness conditions and in the presence of age-related diseases, such as cardiovascular diseases, chronic renal disease, cancer, insulin resistance, and type 2 diabetes [292]. It is also associated with aging independently from the health status of the individuals. This protein is one of the most upregulated proteins during the aging process and a key regulator of the mechanisms of stress response in humans. It has been demonstrated that human GDF15 overexpression in mice extends the median lifespan by 19.5% in transgenic animals and protects against weight gain and insulin insensitivity [293]. It is also associated with several age-related diseases. Many studies indicate that this mitokine has protective roles in different tissues during stress and aging, therefore playing a beneficial role, which contrasts with the observed association in many age-related diseases. GDF15 might then be considered a pleiotropic factor with beneficial activities, but turning detrimental in old age, probably when it is pathologically and chronically augmented. However, it seems to be that the role of GDF15 in the aging process and in age-related diseases is a lot more complex than was thought and is still unclear. It has been proposed that GDF15 might play a role in a dormancy program to mediate tissue tolerance during inflammation and during tissue injury [292]. 

### 13.3. Aging Biomarkers in Different Tissues

A recent systematic review of human proteomics aging studies revealed a novel proteomic aging clock that highlights key biological processes that change with age [290]. The review included 36 different proteomic analyses, which have identified different proteins significantly changing with age. A large set of common proteins (1128) using different proteomic techniques have been reported to change during human aging in two or more different aging analyses, using different tissues and/or cell types. Further, 32 common proteins have been reported by five or more analyses, and these proteins have well-known connections relevant to aging and age-related diseases. When these 1128 commonly identified proteins were subjected to bioinformatics enrichment analysis, it was found that these proteins are implicated in biological processes related to inflammation, extracellular matrix, and gene regulation. Many proteins were unique to a single tissue, as in the case of tumor necrosis factor receptor superfamily member 11B (TNFRSF11B) and IGFBP6, both of which were reported in plasma by four different studies, and both significantly increase with age [294]. Further, there is a tissue overlap for some proteins, for example, the heat shock protein B1 (HSPB1) and Annexin A1 (ANXA1). Of the 1128 common proteins identified, 751 were reported in two or more different tissues and/or cell types. Meantime, the rest (377 proteins) were reported in a single tissue or cell type. Many commonly identified proteins by two or more studies have been previously reported to be connected with aging and summarized in the study [290]. They are APOE, growth differentiation factor 11 (GDF11), insulin-like growth factor 1 (IGF1) signaling pathway, the NAD-dependent histone deacetylase Sirtuin 5 (SIRT5), AKT2 kinase, nicotinamide N-methyltransferase (NNMT), and the related nicotinamide N-methyltransferase enzyme (NAMPT), among several others.

Of these 1128 proteins that were reported to change with age, it was found that three or more studies reported 337, while 117 proteins were reported by four or more different studies, making these the most common proteins identified to change with age. Five more different studies reported only 32 proteins, and from these reported proteins, several can affect life span. Out of these 32 reported proteins, 10 can affect life span in normal and non-diseased animals. These are: ATP synthase subunit beta (ATPF1B), collagen alpha-1 chain (COL1A1), epidermal growth factor (EGFR), fibronectin 1 (FN1), GDF15, glutathione -S-transferase P1 (GSTP1), pro low-density lipoprotein receptor-related protein 1 (LRP1), Parkinson disease protein 7 (PARK7), PTN, and transition protein 1 (TP1). In another related recent study, it has been found that the plasma levels of proteins GDF15, cathepsin S (CTSS), and thrombospondin 2 (THBS2) are significantly associated with mobility disability in the elderly [294]. Moreover, these studies indicate that the identified proteins can be used as high-quality aging biomarkers and furthermore as a good therapeutic target to improve and extend human lifespan. 

### 13.4. Undulating Changes in Biomarkers during Aging

Using the affinity-based slow off-rate modified aptamer (SOMAmer) technology, Lehallier et al. measured 2925 plasma proteins from 4331 healthy males and females ranging from 18 to 95 years old. Using this technology, it was possible to identify undulating proteomic changes during the human lifespan [294]. It was found that waves of changes in the plasma proteome occurred in the fourth, sixth, and eighth decades of life. Differentially expressed plasma proteins during aging presented three local peaks at 34, 60, and 78 years. These waves of changes in the plasma proteome reflect distinct biological pathways and reveal differential associations with the genome and proteome of age-related diseases and phenotypic traits. Females have a longer average lifespan than males, and it was found that sex and aging proteomes are interconnected [294]. Several proteins strongly change with sex, and the most prominent are: follicle-stimulating hormone subunit beta (FSHB), human chorionic gonadotropin subunit beta (CGB), and prostate-specific antigen (KLK3). With age, the most prominent changes, include sclerostin (SOST), ADP ribosylation factor interacting protein 2 (ARFIP2), and GDF15, in addition to several sex proteins. Further, the authors identified a sex-independent plasma proteome clock consisting of 373 proteins that were altered consistently through the life span of the studied individuals. When applied to a machine learning model, this proteomic clock was highly accurate in predicting the biological age of the study participants in both males and females. 

These proteins are grouped in eight clusters with very different trajectories changing with age [294]. Only two groups had linear trajectories, but most had non-linear trajectories changing with age, including stepwise, logarithmic, and exponential ones. Importantly, these cluster trajectories were similarly detectable in four independent cohorts. Out of the eight clusters, six were enriched for specific biological pathways, suggesting distinct, yet orchestrated changes in biological processes occurring during the lifespan. Some interesting observations in this study are as follows. At around 34 years there was a downregulation of proteins involved in structural pathways such as the extracellular matrix. At age 60 years, a prominent role in hormone activity, diverse binding ligands with receptor functions, and blood pathways was found. Finally, at age 78 years, several key processes still included blood pathways and bone morphogenetic protein signaling, which is involved in numerous cellular functions. Altogether, the results show that aging is a dynamic, non-linear process characterized by waves of changes in plasma proteins that reflect complex shifts in biological processes. Furthermore, a comparison to known disease-related protein signatures revealed that the peaks at 60 and 78 years old were enriched for proteins related to Alzheimer’s, cardiovascular diseases, and Down syndrome [294].

## 14. Turning Back the Aging Clocks in Humans by Interventions

Most of the time, chronological age correlates with many age-related diseases and conditions; however, it does not adequately reflect an individual’s capacity, health, or mortality. On the other hand, biological age can provide information about overall health and adequately indicate how rapidly or slowly a person is aging. Biological age can be estimated by aging clocks, which are computational models using a set of input data, such as DNA methylation, plasma or tissue proteins, and transcriptomic data, to make a prediction. Over the last decade, these aging clock studies have revealed that age-related diseases, social variables, and mental health conditions are associated with an increase in predicted biological age compared to chronological age [295]. In addition, biological age acceleration is associated with a higher risk of premature mortality. Recently, studies have reported that the predicted biological age is sensitive to interventions such as caloric restriction, a plant-based diet, a lifestyle involving physical exercises, and drug regimens including metformin [296]. Further, specific molecules such as the antihypertensive drug doxazosin or the metabolite alpha-ketoglutarate have been shown to reduce or reverse the predictive biological age [296]. However, rigorous clinical trials are necessary to validate and use these findings; the current data suggest that aging clocks are malleable and can be changed in humans.

Aging clocks can be turned back in humans according to some trials. For example, the CALERIE trial (Comprehensive Assessment of Long-term Effects of Reducing Intake of Energy) was published in 2015, in which 220 non-obese subjects were randomized and placed on either a 25%-caloric restriction or ad-libitum diet [297]. Although there was only 11.7% mean caloric restriction, this was enough to promote weight loss, decrease circulating TNF-α, and cause a reduction in cardiometabolic risk factors. Using biostatistical methods and clinical biomarkers data that were collected during this trial, the biological ages of individuals in both trial arms were estimated [298]. Ad-libitum and caloric-restricted individuals showed an annual biological age change of 0.71 and 0.11 years, respectively. This delta of 0.6 years was significatively different, and in the ad-libitum group, biological age was significantly higher after 2 years. In conclusion, there was a deceleration of aging in the caloric-restricted group.

In another study, it was reported that dietary interventions can similarly affect an aging clock. A work was conducted on 120 healthy elderly Italian and Polish individuals, who were subjected to a Mediterranean-like diet for 1 year [299], and Horvath classical clock was used to measure epigenetic age in whole blood before and after the 1-year nutritional intervention. The results varied according to sex and country of origin; however, the delta between biological age and chronological age (∆ age) was reduced by 0.84 years in the Polish group. In Polish women, the ∆ age decreased by −1.84 years. These individuals showed an ∆ age that was lower than it was pre-intervention. Differences between the two groups might be due to cultural or social factors. For example, the similarity of a pre-intervention diet to the Mediterranean diet could have influenced the results. A similar study found that an altered diet based on plant foods in a cohort of 219 healthy, postmenopausal women treated for 24 months reduces epigenetic age. ∆ age was 0.64 years lower than that of controls [300]. Further, combination therapies of diet and exercise can lower biological age in healthy individuals [301] and in subjects with obesity or dyslipidemia [302]. In one of the studies [301], a significant decrease in the Horvath clock in treated individuals relative to controls was found. Subjects in the treatment group had an epigenetic age that was 1.96 years lower than when they started the study.

Although it is possible in humans to turn back the aging clock through dietary interventions, aging clocks could also be targeted pharmacologically. The TRIIM (Thymus Regeneration, Immunorestoration, and Insulin Mitigation) trial is a pilot, non-placebo-controlled study, in which a group of 10 healthy adult men (51–65 years old) was treated with metformin, growth hormone, and the protohormone dehydroepiandrosterone (DHEA) [303]. Immunological changes were found, and the epigenetic age was reversed after 1 year of treatment. The subjects had an epigenetic age that was 1.5 years younger than they started the treatment 1 year earlier according to the GrimAge clock [303]. Vitamin D3 supplementation in obese/overweight individuals with low vitamin D also decreased epigenetic age (measured by Horvath or Hannum clock) by 1.85 or 1.90 years compared to placebo-treated individuals [304]. 

Taken altogether, the findings from these studies indicate that it is possible to intervene and decrease biological age in humans; however, we should be aware that these trials are preliminary and short-term using a relatively small number of subjects. Large-scale placebo-controlled studies are needed to validate these results. These trials should be long-term and determine the extent to which biological age changes by using more than one aging clock. 

## 15. Concluding Remarks

Aging is a complex biological process, affecting mainly multicellular organisms, which has been intensively studied in the last three decades. Nine hallmarks of aging have been defined, which can serve as a framework for the upcoming studies on the mechanisms of aging. These hallmarks are all interconnected with each other, and their contributions to the aging process have been documented. Further, these studies might help to design strategies to extend the lifespan and improve the health span of human beings. Nevertheless, there are still many questions that remain to be answered to understand this complex biological process deeply. It is expected that in the future a combination of molecular biology, molecular genetics, cellular biology, next generation sequencing, and multi-OMICS approaches will help to understand the underlying mechanisms of aging and age-related diseases. Today, it is possible to determine the whole-genome sequence of longevous individuals to determine genetic and epigenetic changes with high resolution. Further studies in animal models are also important to provide evidence and assay the contributions of each individual hallmark and to compare short- and long-lived strains and analyze the genetic and epigenetic changes produced. 

Nowadays, biomarkers are obtained from OMICS studies, and with the help of machine-learning approaches, the obtained data can be used to construct highly accurate aging clocks to predict biological age. Some of these biomarkers can be used as a target for therapeutic interventions to improve the health span and extend the human lifespan. Trial assays will be necessary to assay diverse strategies to turn back the biological clocks. These trial assays should be performed with a representative population, and more than one aging clock must be used to determine the biological age. Hopefully, future studies will be able to unravel the underlying mechanisms of aging and will facilitate interventions to improve the health span and extend longevity.

## Figures and Tables

**Figure 1 antioxidants-12-00651-f001:**
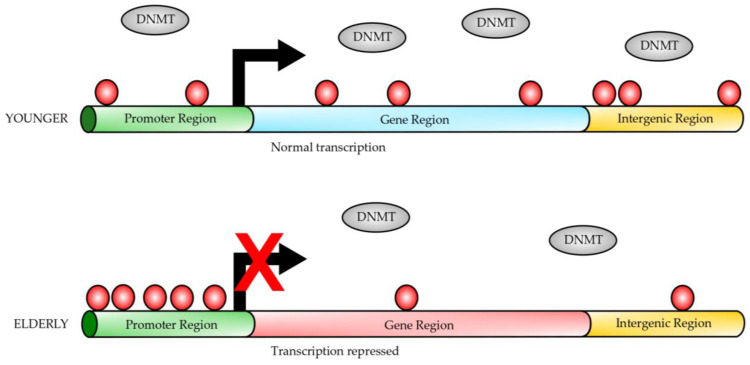
Gene DNA methylation during aging. In younger individuals, the gene promoters have a lower degree of DNA methylation (red circles) than in elderly individuals, while the gene bodies are highly methylated (red circles). On the contrary, elderly individuals have gene promoters heavily methylated (red circles); however, the gene bodies are hypomethylated. Transcription initiation is inhibited by DNA methylation at the gene promoters from elderly individuals. Dynamic DNA methylation occurs at CpG sites on the gene promoters or gene bodies. DNA methylases (DNMTs) carry out methylation of CpG sites, which protein factors or non-coding RNAs can regulate. This figure was modified from reference [21].

**Figure 2 antioxidants-12-00651-f002:**
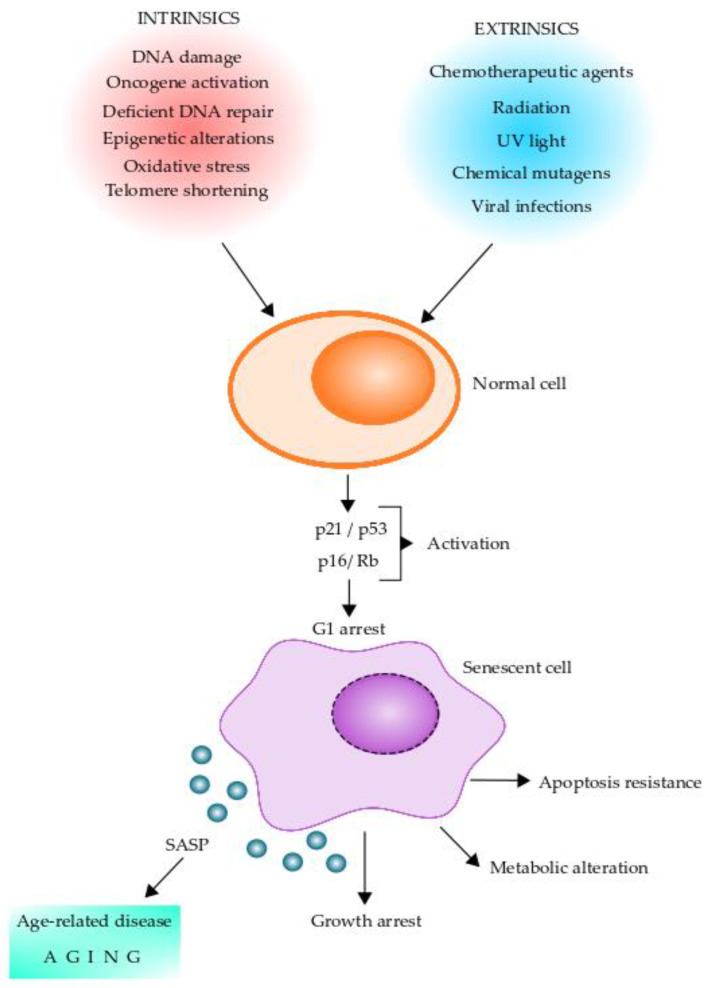
Stress stimuli leading to cellular senescence. Senescent cells can play pleiotropic roles in normal physiology, pathology, and aging. Diverse factors (intrinsic and extrinsic) can act on normal cells and induce a senescent phenotype. Senescent cells initiate the senescent cellular program by activating the p16/Rb and/or p21/p53 tumor suppressor pathways, which can arrest the cells in G1. Senescent cells undergo several intracellular changes, which are shown in the figure. One of the main features of senescent cells is the production of the senescence-associated secretory phenotype (SASP), which contains a series of molecules that can alter neighboring cells leading to age-related diseases and aging. Intracellular changes of senescent cells include DNA-SCARS (DNA segments with chromatin alterations reinforcing senescence), CCF (cytoplasm chromatin fragments), SADF (senescent-associated DNA damaged foci), SAHF (senescent-associated chromatin foci), SAMD (senescent-associated mitochondrial dysfunction), mtDNA (mitochondrial DNA), and TAF (telomere-associated foci).

**Figure 3 antioxidants-12-00651-f003:**
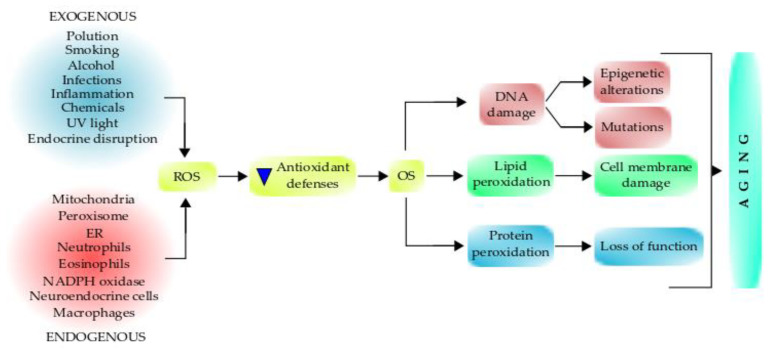
Role of free radicals on oxidative stress and the aging process. Various endogenous and exogenous factors can generate free radicals/oxidants, affecting the antioxidant defenses, which promote oxidative stress that can damage different biomolecules. The damaged biomolecules affect inter and intracellular signaling and gene expression pathways, causing several cell alterations. The interplay between these events can lead to aging, which increases aging-associated syndromes and aging-related diseases. Antioxidant supplementation can retard the aging process.

**Figure 4 antioxidants-12-00651-f004:**
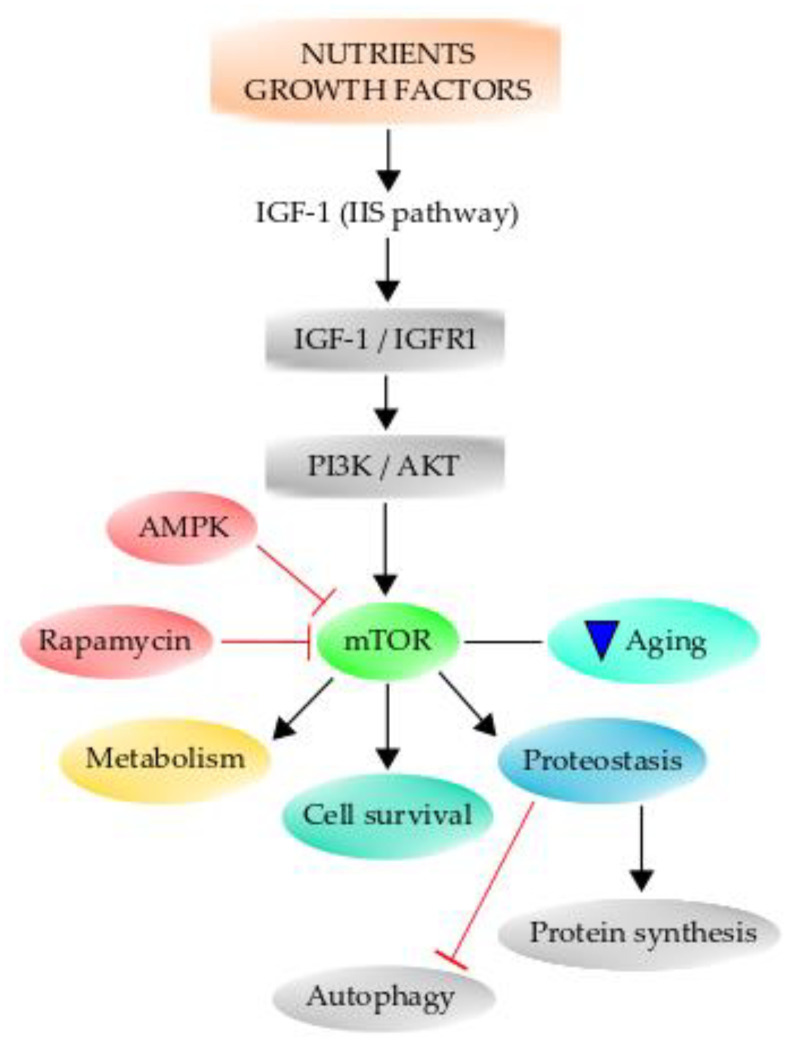
The mammalian TOR pathway (mTOR) can regulate several hallmarks of aging. Growth factors and nutrient availability can activate the IIS pathway to regulate mTOR, which acts on downstream targets, regulating metabolism, cell survival, and proteostasis, among several others. mTOR inhibition by rapamycin or by AMPK can decrease aging. Further, mTOR is able to negatively regulate autophagy and positively regulate protein synthesis.

**Figure 5 antioxidants-12-00651-f005:**
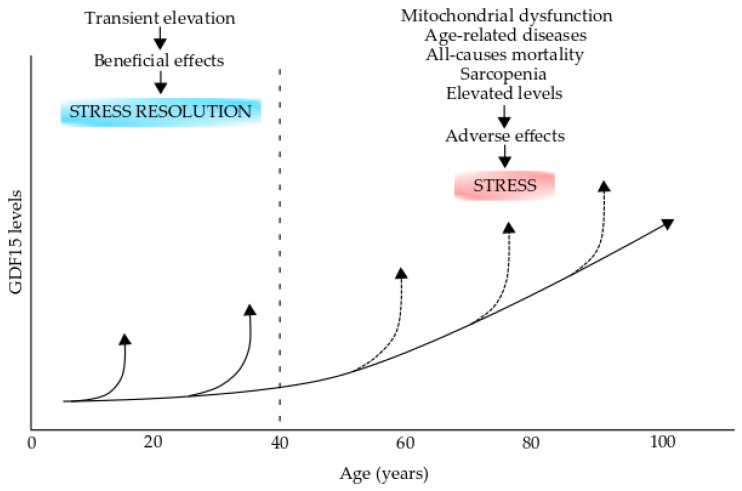
Growth differentiation factor 15 (GDF15) levels in aging and aging-related diseases. The GDF15 levels increase during the process of normal aging. GDF15 transient elevations during early life (up to 40 years old) could have beneficial effects on stress resolution However, these levels can increase over the normal levels (broken arrows) in individuals over 40 years old and could have adverse effects and produce stress. Elevated levels are produced during mitochondrial dysfunction, age-related diseases, all-cause mortality, and sarcopenia. This figure was modified from reference [269].

## Data Availability

All data are shown within the manuscript.

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
