# Peer review of "Aging Hallmarks and the Role of Oxidative Stress"

_antioxidants, 2023, doi:10.3390/antiox12030651_

Round 1

Reviewer 1 Report

Thank you for the opportunity to review this manuscript, dealing with interesting explanations entitled “The aging hallmarks and the role of oxidative stress”. In this article, the author has focused on nine hallmarks of the aging process, including shortening telomere length, genomic instability, epigenetic modifications, mitochondrial dysfunction, loss of proteostasis, dysregulated nutrient sensing, stem cell exhaustion, cellular senescence, and altered cellular communication. In this review, they summarize recent advances in the mechanisms of aging with an emphasis on mitochondrial dysfunction and ROS production. Certainly, these alterations have been linked to sustained systemic inflammation and mitochondrial dysfunction is one of the most important mechanisms contributing to the aging process. However, this manuscript needs to be substantially revised with the part that they should focus on as per their statement in the abstract in addition to the novelty in all Figures before submitting it to this journal.

·         As per the author’s statement, this review should focus on recent advances in the mechanisms of aging with an emphasis on mitochondrial dysfunction and ROS production. However, mitochondrial dysfunction and ROS production's role in the aging process are not well documented in the manuscript. The author needs to add at least one table and 1-2 figures that show the mechanistic explanation of mitochondrial perturbation/stress in the aging process. 

·         There’s nothing new in Figure 1. The replicate overview of dynamic DNA methylation during aging is already available (Figure 1, Xiao et al. Front. Genet., 10, 2019). If the author gets an idea from any published Figure needs to substantially modify it before submitting, it to any journal as per copyright agreement, or the author needs to get permission from the published journal to include it in their publication if it is necessary to support their hypothesis.

·         Figure 2 shows in what way stress stimuli lead to cellular senescence. Unfortunately, it is also an exact copy of another figure that’s from a recently published review (Zhang et al. J Clin Invest. 2022;132(15):e158450). The author needs to check the whole manuscript for plagiarism before submitting it again.

·         Figure 3 illustrates the role of free radicals on oxidative stress and the aging process. This figure shows how various endogenous and exogenous factors affect the aging process by inducing cellular stress. I don’t know why the author included this figure in the “Mitochondrial dysfunction plays a crucial role in aging” section. This section must include any table of figures which only confirms how mitochondrial dysfunction (free energy change, functional enzymes reduction, glycolysis imbalance, or mitochondrial free radicals involved in aging. Therefore, Figure 3 does not fit in this position/material.

·         Nothing new has occurred in Figure 4 as well. This figure is an exact copy of another flow chart published by Papadopoli et al (2022, Figure 2) entitled “mTOR as a central regulator of lifespan and aging.”

·         Likewise, Figure 5, which represents the growth differentiation factor 15 levels in aging and aging-related diseases, is also a copy of the figure of the recently published book chapter entitled “Mitochondrial Dysfunction and Growth Differentiation Factor 15 in Aging” by Fujita and Tanaka (Aging Mechanism II, pp 157–173, 2022).

Author Response

Below, we have enclosed the answers to the reviewers’ comments. We would like to thank to all three reviewers for their helpful comments on the manuscript. Those comments will certainly improve the quality of the whole manuscript. We expect that now the manuscript could be suitable for publication in the Antioxidants journal.

Reviewer 1

  1. We have expanded the section on ROS production and mitochondrial dysfunction. We added an original figure (Fig 3) to illustrate the whole process of ROS production and oxidative stress and its consequences on the aging process.
  2. We believe that figure 1 is necessary to illustrate the process of dynamic gene DNA methylation and how it influences aging. That figure has already been redrawn, and permission has been obtained from the publisher.
  3. Figure 2 was replaced by an original figure from us. We have checked the whole manuscript for any plagiarism, and we eliminated any plagiarism found throughout the whole manuscript.
  4. Figure 4 was eliminated and replaced with a figure from us.
  5. Figure 5 was completely modified and redrawn. Moreover, we have indicated the source of this new figure.

Reviewer 2 Report

The review is dedicated to the aging hallmarks in aging, including epigenetic modifications, signaling alterations, mitochondrial disfunction and stem cells senescence. The study contains a lot of examples and references to the research studies concerning aging-associated processes with the research objects from humans to c elegans. However, most of these examples are associated with neuronal tissue and neurodegenerative diseases, so the authors should concern changing the title of the review to specify the “neuronal” focus of the study.

There is also a number of some inaccurate points in this study, for example, the authors write that “Dnmt1 160 maintains the methylation patterns in the genome by replicating the hemimethylated CpG 161 sites [32], while Dnmt3a and Dnmt3b are considered de novo DNA methylases”. In fact, Dnmt1 participates in methylating mitochondrial DNA and oxidative metabolism-related promoters in an AMPK-dependent manner (for review see F. C. Lopes, A. Mitochondrial metabolism and DNA methylation: a review of the interaction between two genomes. Clin Epigenet 12, 182 (2020). https://doi.org/10.1186/s13148-020-00976-5,    Sharlo, K.A., Lvova, I.D. & Shenkman, B.S. Interaction of Oxidative Metabolism and Epigenetic Regulation of Gene Expression under Muscle Functional Unloading. J Evol Biochem Phys 58, 625–643 (2022). https://doi.org/10.1134/S0022093022030012)

The authors also write that “ROS is produced by electron leakage from complex I and complex III from electron transport chains leading to a partial reduction of oxygen to form superoxide”, however, these complexes are not the only sources of mitochondrial ROS – see Mailloux RJ. An Update on Mitochondrial Reactive Oxygen Species Production. Antioxidants. 2020; 9(6):472. https://doi.org/10.3390/antiox9060472.

There are also some lacks of information in the topics about CpG-methylation and proteolysis. Speaking of CpG-methylation one should mention ten-eleven translocases and their role in agening-induced changes (see Truong TP, Sakata-Yanagimoto M, Yamada M, Nagae G, Enami T, Nakamoto-Matsubara R, Aburatani H, Chiba S. Age-Dependent Decrease of DNA Hydroxymethylation in Human T Cells. J Clin Exp Hematop. 2015;55(1):1-6. doi: 10.3960/jslrt.55.1. PMID: 26105999. Gontier G, Iyer M, Shea JM, Bieri G, Wheatley EG, Ramalho-Santos M, Villeda SA. Tet2 Rescues Age-Related Regenerative Decline and Enhances Cognitive Function in the Adult Mouse Brain. Cell Rep. 2018 Feb 20;22(8):1974-1981. doi: 10.1016/j.celrep.2018.02.001. PMID: 29466726; PMCID: PMC5870899.

Speaking of age-dependent proteolysis alterations needs mentioning age-related sarcopenia, its causes, mechanisms and consequences. I hope that adding of this information would straighten the review.

Author Response

Below, we have enclosed the answers to the reviewers’ comments. We would like to thank to all three reviewers for their helpful comments on the manuscript. Those comments will certainly improve the quality of the whole manuscript. We expect that now the manuscript could be suitable for publication in the Antioxidants journal.

Reviewer 2

We do not completely agree to change the manuscript title since we intend to provide a general view of the aging hallmarks with an emphasis on ROS production and mitochondrial dysfunction. Besides, several excellent reviews have been published on aging and neurodegenerative diseases.

  1. The information on the role of Dnmts and mitochondrial DNA methylation has been included in the manuscript text.
  2. Additional mitochondrial ROS sources have been included in the text.
  3. The role of ten eleven translocases has been included and cited in the manuscript text.
  4. A new section on sarcopenia was included. Note that sarcopenia mechanisms are not completely understood yet, and that has been indicated in the text.

Reviewer 3 Report

It is a well-written manuscript. The reviewer only has some minor comments.

 Line 297-298 “Remarkably, the transplantation of a small number of senescent cells into young healthy animals recapitulates age related impaired physical function [88, 89]”.

Do authors have opposite results to further support this comment. For example, could transplantation of a small number of young cells into aged animals decreases the age related impaired physical function?

Line 565 “Mitochondrial dysfunction plays a crucial role in aging”

Authors discussed “Proteostasis dysfunction in the aging process” in section 7. Recent studies show that ER stress plays key role in mitochondrial dysfunction during aging. Could Authors discuss the role of ER stress in mitochondrial dysfunction here.

Author Response

Below, we have enclosed the answers to the reviewers’ comments. We would like to thank to all three reviewers for their helpful comments on the manuscript. Those comments will certainly improve the quality of the whole manuscript. We expect that now the manuscript could be suitable for publication in the Antioxidants journal.

Reviewer 3.

  1. Yes, transplantation of a small number of young cells into aged animals can decrease age-related impaired physical function, as has been noted in the manuscript text.
  2. The role of ER stress in mitochondrial dysfunction has been included and cited from studies that have been performed in mitochondria from aged hearts.

Round 2

Reviewer 1 Report

The revised manuscript and the author's response are satisfactory. Therefore, the manuscript can be considered for publication in the Antioxidants journal in its present form.

Thank you

Reviewer 2 Report

The study has been significantly improved in the revised version